# Structural Optimization and Mechanical Simulation of MEMS Thin-Film Getter–Heater Unit

**DOI:** 10.3390/mi13122252

**Published:** 2022-12-18

**Authors:** Xinlin Peng, Yucheng Ji, Shuo Chen, Song Guo, Liuhaodong Feng, Yang Xu, Shinan Wang

**Affiliations:** 1School of Microelectronics, Shanghai University, Shanghai 201800, China; 2Shanghai Industrial μTechnology Research Institute, Shanghai 201899, China

**Keywords:** MEMS getter–heater unit, cantilever model, symmetrically-shaped heater and edge–center-located cantilever structure, stress and deformation displacement simulation, interfacial stress simulation

## Abstract

A MEMS thin-film getter–heater unit has been previously proposed for the vacuum packaging of a Micro-Electro-Mechanical System (MEMS) device, where the floating structure (FS) design is found to be obviously more power-efficient than the solid structure (SS) one by heat transfer capacity simulation. However, the mechanical strength of the FS is weaker than the SS by nature. For high temperature usage, the unit structure must be optimized in order to avoid fracture of the cantilever beam or film delamination due to strong excessive stress caused by heating. In this paper, COMSOL is used to simulate the stress and deformation of the MEMS thin-film getter–heater unit with the cantilever structure. By comparing various cantilever structures, it is found that a model with a symmetrically-shaped heater and edge–center-located cantilever model (II-ECLC model) is the most suitable. In this model, even when the structure is heated to about 600 °C, the maximum stress of the cantilever beam is only 455 MPa, much lower than the tensile strength of silicon nitride (Si_3_N_4_, 12 GPa), and the maximum deformation displacement is about 200 μm. Meanwhile, the interfacial stress between the getter and the insulating layer is 44 MPa, sufficiently lower than the adhesion strength between silicon nitride film and titanium film (400–1850 MPa). It is further found that both the stress of the cantilever structure and the interfacial stress between the getter and the insulating layer beneath increase linearly with temperature; and the deformation of the cantilever structure is proportional to its stress. This work gives guidance on the design of MEMS devices with cantilever structures and works in high temperature situations.

## 1. Introduction

In recent years, the application of MEMS devices is ubiquitous. High-end MEMS devices, such as MEMS accelerometers and gyroscopes with high-speed vibration structures, usually require a stable high vacuum environment to maintain their performance. For example, a MEMS gyroscope must be packaged in a sealed cavity with a high vacuum degree to avoid the air damping effect which decreases the quality factor (Q) and affects the performance of the device [1]. In addition, there are reports addressing the importance of a high vacuum in maintaining the performance of MEMS pressure sensors and micromirrors [2,3,4]. To achieve the requirement of a high degree of vacuum in the sealed cavity, getter technology is generally used. The getter absorbs the gases enclosed in the cavity during packaging, released by the materials in the sealed cavity.

The non-evaporable getters can be activated multiple times at low temperature without particle contamination to the device [5]. In practice, the non-evaporable getters are manufactured with a passivation layer on the surface to prevent damage by subsequent processing or transportation. In addition, the non-evaporable getters need to be heated continuously under vacuum and high temperature before they can work. During this process, known as getter activation [5,6,7], the surface passivation layer diffuses into the bulk, exposing the active surface in order for the getter to efficiently absorb the gas. Knight et al. [8,9] found that the SAES getter needs to be heated at 425 °C for 30 min for sufficient activation. Ge et al. [10] developed a getter that can be activated at 150 °C, but for up to 12 h. According to those reports, increasing the activation temperature can shorten the activation time. However, the packaging temperature of the device will also be increased, which affects the structure and performance of the device.

As an alternative solution, we proposed a thin-film getter–heater unit, studied its heating activation performance and preferred a floating structure [11]. The floating structure is commonly used in MEMS devices to reduce heat loss. In common applications, the working temperature of the floating structure is generally below 200 °C. For example, the working temperature of an accelerometer is in the range of −50 °C up to 120 °C [12], and that of a MEMS Pirani is about 100 °C [13]. However, for the getter–heater unit mentioned above, the floating structure (FS) model needs to be heated above 200 °C, or even 600 °C, to ensure that the getter can be activated sufficiently and efficiently. In this paper, we optimize the structure of the FS model from the perspective of thermal stress and thermal expansion to ensure that the beam will not break due to stress and the membrane will not be delaminated due to deformation at high temperatures. The material of the cantilever structure is silicon nitride. The structure of the FS model is optimized by exploring the thermal stress and thermal expansion of different cantilever structures using the finite element software COMSOL. Furthermore, the reliability of the adhesion between the cantilever structure and the titanium-based thin film getter is evaluated.

## 2. Geometric and Physical Model Construction

### 2.1. Geometric Model Construction

Based on the floating model in the previous study [11], the structure of the insulating layer is simplified to obtain the edge–edge-located cantilever model (abbreviated as the EELC model), as shown in Figure 1. The whole simulation model is divided into 6 parts: namely, substrate, cantilever structure, electrode, heater, insulating layer and getter.

The parameters of the geometric structure are shown in Table 1. According to the size of the common gyroscope (about 2–5 mm), we designed the getter–heater size of 1800 × 1800 μm. The cavity depth is set to 300 μm, which can be fully released with a KOH solution. In order to reduce the heat conduction to the substrate, we designed the cantilever beam with the shape of an edge cantilever and also determined the distribution shape of the wire. From practical experience, we designed the width of the wire to be 50 um and the thickness to be 0.2 μm. Through the simulation of the number of wires in the early stage [11], the final interval between wires is 100 μm. Considering the processing capacity of our PECVD machine, the thickness of the cantilever beam is determined to be 1.5 μm. Given that the width of the cantilever beam needs to be larger than the width of the wire, the width of the cantilever beam is set to 100 μm. The insulating layer needs to cover the entire wire, at least twice the thickness of the wire, so we choose 0.4 μm. Finally, the thickness of getter film is determined according to its air suction performance. The main parameters of the cantilever structure are $d_{b}$ and $t_{b}$; $d_{b}$ represents the width of the cantilever beam and $t_{b}$ represents the thickness of the cantilever structure, as shown in Figure 1.

### 2.2. Physical Model Construction

The thermodynamic simulation of the MEMS thin-film getter–heater unit is mainly to analyze the stress of the cantilever structure and the interfacial stress between the insulating layer and the getter at high temperature. In the simulation model, the energy balance relationship in solid heat conduction, the relationship between thermal stress and temperature in thermodynamics and the relationship between stress and strain in elastic mechanics are required. The model is simplified by assuming that the materials in the model are linear elastic materials.

#### 2.2.1. Solid Heat Conduction

The heat in the floating structure comes from the Joule heat of the resistance. According to the Fourier conduction theorem, the heat conduction equation is:(1)$$\rho C_{P}\frac{\partial T}{\partial t}=Q$$
where ρ is the density of the material (kg/m^3^), $C_{P}\;$is the specific heat capacity of the material (J/kg·K), T is the temperature of the material (K) and Q is the Joule heat generated by the resistance (W/m^3^).

When the model conducts heat, the resistance heat cannot be completely transferred to the floating structure. Since the getter–heater unit works in a vacuum environment, heat convection can be ignored, and the radiant heat $Q_{r}$ (W/m^3^) needs to be considered. At the same time, the energy $W_{\sigma }$ (W/m^3^) dissipated by the conduction thermal resistance needs to be considered in the process of heat transmission. Thus, the heat conduction equation of the model is:(2)$$\rho C_{P}\frac{\partial T}{\partial t}=Q+Q_{r}+W_{\sigma }$$

According to the Stefan–Boltzmann law, the calculation formula for radiant heat is:(3)$$Q_{r}=\nabla \cdot q_{r}$$where $q_{r}\;$is the radiation heat flux (W/m^2^), and the calculation formula of $q_{r}\;$is:(4)$$q_{r}=-k\nabla T$$

Since the thermal expansion coefficient α (1/K) of the linear elastic material is a fixed value, the heat loss of the conduction thermal resistance is defined as:(5)$$W_{\sigma }=\alpha T:\frac{\partial S}{\partial t}\;$$
where S is the stress tensor (Pa). “:” refers to double-dot multiplication, which means that the components are multiplied in a certain order (refer to [14]). Combining Equations (2)–(5), the final heat conduction equation of the model is obtained as:(6)$$\rho C_{P}\frac{\partial T}{\partial t}=Q-\nabla \cdot \left(k\nabla T\right)+\alpha T:\frac{\partial S}{\partial t}$$

#### 2.2.2. Solid Thermodynamics

The model is subjected to force analysis, the solid deformation force is derived from the sum of the external and internal forces on the solid and the mechanical equation is obtained as:(7)$$\rho \frac{d^{2}u}{dt^{2}}=\nabla \cdot S+F_{v}$$
where S is the stress tensor (Pa), $F_{v}\;$is the external body force (N/m^3^) and u is the deformation displacement (m). Stress and strain satisfy Hooke’s law:(8)$$S=C:\varepsilon_{el}$$
where C is the elastic tensor (Pa), ε is the strain tensor and $\varepsilon_{el}$ is the elastic strain tensor. Under the condition of ignoring the nonlinear deformation of the material and considering the relationship between thermal strain and temperature, the relationship that the strain satisfies is:(9)$$\varepsilon_{el}=\left(\varepsilon -\varepsilon_{th}\right)$$
(10)$$\varepsilon_{th}=\alpha \left(T-T_{ref}\right)$$
where $\varepsilon_{th\;}$is the thermal strain, T is the structure temperature (K), the volume reference temperature of the $T_{ref}$ structure (293.15 K), and the transformation relationship between the strain tensor and the deformation displacement is:(11)$$\varepsilon =\frac{1}{2}\left[{\left(\nabla u\right)}^{\prime }+\nabla u\right]$$
where “′” represents the transpose of the matrix.

Combining Equations (7)–(11), the final solid thermodynamic equation of the model is obtained as:(12)$$\rho \frac{d^{2}u}{{\mathrm{dt}}^{2}}=\nabla \cdot S+F_{v}$$
(13)$$S=C:\left\{\frac{1}{2}\left[{\left(\nabla u\right)}^{\prime }+\nabla u\right]-\alpha \left(T-T_{ref}\right)\right\}$$

## 3. Model Simulation and Optimization

### 3.1. The Edge–Edge-Located Cantilever Model

In the previous work [11], in order to improve the heat transfer efficiency of the integrated unit, the length of the cantilever beam was increased, and the edge–edge-located cantilever model (EELC model) was selected, as shown in Figure 1. In this paper, the mechanical simulation of the floating model is mainly carried out. The main parameters of the floating model in the simulation are shown in Table 1. Before the simulation, it was stipulated that the deformation to the side of the cavity is downward, which is taken as negative; otherwise, it is upward, which is taken as positive. The EELC model is simulated, when the voltage of 2.35 V is applied, the maximum temperature of the cantilever structure reaches 600 °C. The simulation cloud diagram is shown in Figure 2; the maximum stress of the cantilever structure is 424.84 MPa, and the maximum deformation displacement is −279.27 μm. The deformation of the entire cantilever structure is downward, and the maximum deformation displacement is at the turning point of the cantilever beam. Since the cantilever beam of the EELC model is long, the rigidity of the cantilever beam is poor, and the stress concentration at the turning point is considerable. Although the upper insulating layer and the getter are heated and deformed upward, the deformation displacement is smaller than that at the turning point of the cantilever beam, so the deformation displacement of the entire cantilever structure is downward. This large downward deformation displacement may cause the cantilever structure to contact the bottom of the cavity during heating, resulting in partial heat loss and an inability to fully activate the getter in the upper layer; thus, the structure needs to be optimized.

### 3.2. The Corner–Corner-Located Cantilever Model

Based on the design concept of the previous structure, the model is improved, and the corner–corner-located cantilever model (abbreviated as the CCLC model) is proposed, as shown in Figure 3. The CCLC model can reduce the length of the cantilever beam and enhance the mechanical properties of the cantilever structure. It is found that the maximum temperature of the cantilever structure can reach 600 °C when an input voltage of 3.7 V is applied, and the heating performance is significantly lower than that of the EELC model. The simulation cloud diagram is shown in Figure 4. The deformation of the cantilever structure is downward, and the maximum deformation displacement is −112.19 μm, located in the geometric center. Moreover, the maximum stress of the cantilever structure is 449.13 MPa. Compared with the EELC model, the cantilever beam of the CCLC model is shorter. It conducts more heat to the substrate and leads to a larger heat loss, resulting in a decrease in heat transfer performance. In this model, the rigidity of the cantilever beam is high, and the deformation of the cantilever beam is downward which is greater than that at the geometric center of the cantilever structure. Thus, the entire cantilever structure is deformed downward.

### 3.3. The Edge–Center-Located Cantilever Model

Since the cantilever beam of the CCLC model is short, the heat loss increases significantly, and the model needs to be further optimized. Considering the advantages and disadvantages of the previous two models, the type I edge–center-located cantilever model (abbreviated as the I-ECLC model) is proposed, as shown in Figure 5a. When the input voltage is 2.28 V, the maximum temperature of the cantilever structure can reach 600 °C. Compared with the EELC model, the heat transfer performance of the I-ECLC model is slightly better. Given that the model has a shorter cantilever beam, the heat loss caused by the cantilever beam is more. However, as the metal lines of the heater become shorter and the resistance value is reduced, the input voltage required to reach the same temperature becomes smaller. The simulation cloud diagram is shown in Figure 6. The maximum stress of the cantilever structure is 488.13 MPa, and the maximum deformation displacement is +197.29 μm. In addition, the deformation of the entire cantilever structure is upward, and the maximum stress and deformation are located at its geometric center. Compared with the EELC model and CCLC model, the direction of the deformation displacement has changed significantly. This is because the upward deformation displacement of the insulating layer and the getter is significantly larger than the downward deformation displacement of the cantilever beam, resulting in the deformation displacement of the entire cantilever structure being upwards.

The effective use of metal lines is considered in the design of the I-ECLC model. The distribution of metal lines is not completely symmetrical, resulting in uneven temperature and stress on the getter, which affects the activation efficiency. The structure of the heater needs to be optimized, and the type II edge–center-located cantilever model (abbreviated as the II-ECLC model) is proposed, as shown in Figure 5b. When the input voltage is 2.27 V, the maximum temperature of the model can reach 600 °C. The simulation cloud diagram is shown in Figure 7. The maximum stress of the cantilever structure is 455.40 MPa, and the maximum deformation displacement is +199.96 μm. Compared with the I-ECLC model, the distribution of temperature and stress are more uniform, the area range of the high temperature increases and the maximum stress decreases to a certain extent. The increase in the symmetry and area of the heater structure in this model not only makes the stress of the cantilever structure more uniform but also improves the activation efficiency of the entire getter. According to reference [15], the tensile strength of silicon nitride is 12 GPa, which is much larger than the stress of the cantilever structure, so there is no cantilever fracture failure.

## 4. Thermodynamic Simulation and Analysis of II-ECLC Model

### 4.1. Simulation of Stress and Deformation Displacement of Cantilever Structure

Increasing the input voltage ($V_{in}$), the highest temperature ($T_{b})\;$of the cantilever structure, the stress ($S_{b}$) and the deformation displacement ($Z_{b}$) in part of the cantilever structure at $T_{b}\;$are obtained; the relationship between the three is studied. The simulation results are shown in Figure 8. As the input voltage increases, $T_{b}$, $S_{b}$ and $Z_{b}$ increase continuously. In addition, the rate of change gradually increases, showing a quadratic relationship. It is not only further proof of the previous study, but also shows that $T_{b}$ and $Z_{b}$ are related to $V_{in}$:(14)$$S_{b}=4.05V_{in}^{2}+5.6V_{in}+18.11$$
(15)$$Z_{b}=2.47V_{in}^{2}+3.49V_{in}+10.81$$

The temperature of the cantilever structure increases, resulting in thermal stress, which causes the deformation of the cantilever structure. The higher the temperature, the greater the stress. In addition, $t_{b}$ and $d_{b}$ are important geometric parameters of the cantilever structure. The relationship of $T_{b}$, $S_{b}$ and $t_{b}$ and $d_{b}$ is shown in Figure 9 and Figure 10. Obviously, $Z_{b}$ and$\;T_{b}$ are linearly related to $S_{b}$. When $t_{b}$ or $d_{b}$ increases, $S_{b}$ and $Z_{b}$ decrease. Moreover, the increase of $d_{b}$ does not affect the changing rate of stress that is dependent on temperature; however, it leads to low heat transfer efficiency. When $t_{b}$ or $d_{b}$ are reduced, the temperature of the cantilever structure rises significantly, causing large stress and deformation displacement, which indicates poor mechanical properties under the same input voltage. Finally, the optimal structural parameters of the cantilever structure are chosen ($t_{b}$ is 1.5 μm and $d_{b}$ is 100 μm) by considering the mechanical strength performance and heat conduction efficiency comprehensively.

In this model, $S_{b}$ and $T_{b}$ are linearly related:(16)$$S_{b}=0.56T_{b}-11.22$$

$Z_{b}$ has a linear relationship with $S_{b}$ of the cantilever structure, as shown in Figure 10, and the fitted equation is:(17)$$Z_{b}=0.61S_{b}-0.14$$

As the temperature increases, the stress increases linearly, and the deformation displacement increases linearly with the stress, which indicates that the floating model conforms to the relationship between stress and temperature in thermodynamics and the generalized Hooke’s law.

### 4.2. Simulation and Analysis of Interfacial Stress between Insulating Layer and Getter

In the II-ECLC model, with a $t_{b}$ of 1.5 μm and a $d_{b}$ of 100 μm, a voltage of 2.27 V is applied, and the maximum temperature reaches about 600 °C. At the same time, the maximum value of the interfacial stress ($\sigma_{b}$) between the insulating layer and the getter is 44.04 MPa. The interfacial stress ($\sigma_{b}$) obtained by simulation is shown in Figure 11. The experience and related references [16,17,18,19] show the adhesion force is 400–1850 MPa between the two materials, which is significantly greater than the interfacial stress, so the separation phenomenon does not occur. In order to simplify the simulation model, the solid material is used for the thin film getter in this simulation. However, in practical applications, the getter is generally a porous material, and the interfacial strength will be slightly lower. Therefore, it is necessary to further explore the relationship between $T_{b}\;$and $\sigma_{b}\;$to provide a theoretical reference for the design and optimization of floating devices. The simulation shows that the higher the temperature, the greater the interfacial stress between the insulating layer and the thin film getter, which is linearly increased. As shown in Figure 12, the fitted relationship is:(18)$$\sigma_{b}=0.076T_{b}-1.59$$

## 5. Conclusions

In this paper, the floating structure model of the thin-film getter–heater unit has been deeply analyzed based on the finite element method in order to avoid structure breakage or too-large deformation and film delamination by thermal stress at high temperatures. Four structural models are established. One conclusion is that the shape and location of the supporting cantilevers are a key factor that affects the thermal stress and thermal deformation of the floating structure model. The edge–center-located cantilever model has smaller thermal deformation and an upward deformation trend, compared with the edge–edge-located cantilever model. Furthermore, the temperature distribution uniformity of the type II edge–center-located cantilever model (II-ECLC model) is better due to the symmetrical distribution of the metal lines, compared with the type I edge–center-located cantilever model (I-ECLC model). The stress of the II-ECLC model at 600 °C is 455 MPa, which is significantly lower than the tensile strength of silicon nitride (12 GPa). The simulation results show that the maximum thermal stress of the cantilever structure is proportional to the temperature, and its deformation is linearly related to the thermal stress. In addition, the analysis showed that the interfacial stress between the getter and the silicon nitride layer is 44 MPa at 600 °C, much lower than their adhesion strength (400–1850 MPa), meaning that there is not the risk of film delamination even at such a high temperature. The results in this paper give guidance for the design of MEMS devices with cantilever structures.

## Figures and Tables

**Figure 1 micromachines-13-02252-f001:**
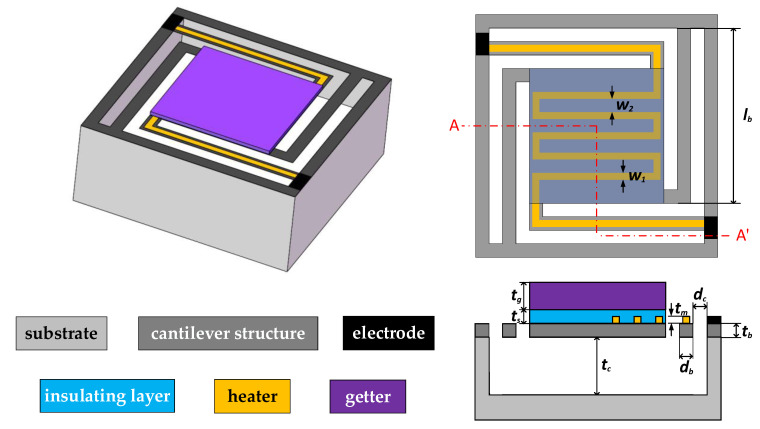
EELC model.

**Figure 2 micromachines-13-02252-f002:**
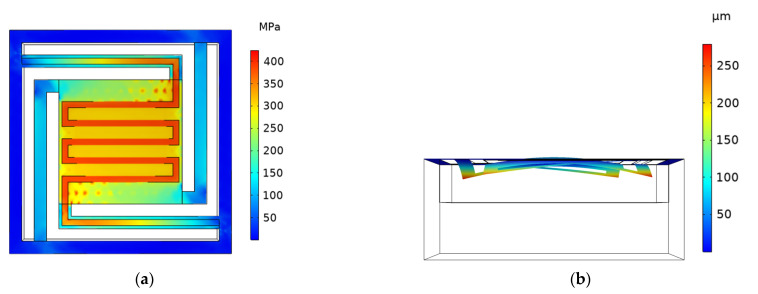
Cloud diagram of stress and deformation displacement simulation of the EELC model. (**a**) Cloud diagram of stress simulation; (**b**) cloud diagram of deformation displacement simulation.

**Figure 3 micromachines-13-02252-f003:**
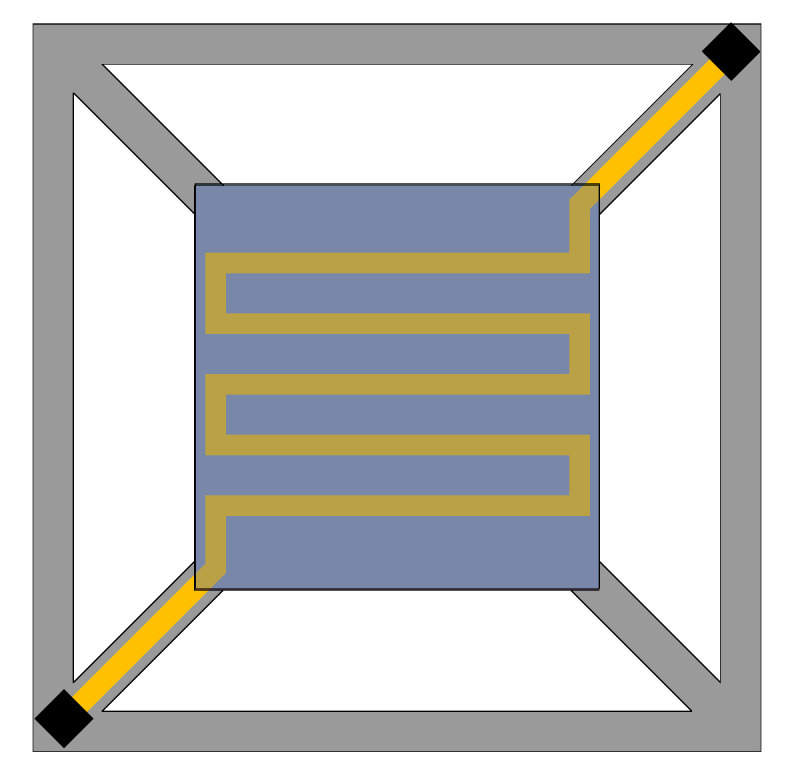
CCLC model.

**Figure 4 micromachines-13-02252-f004:**
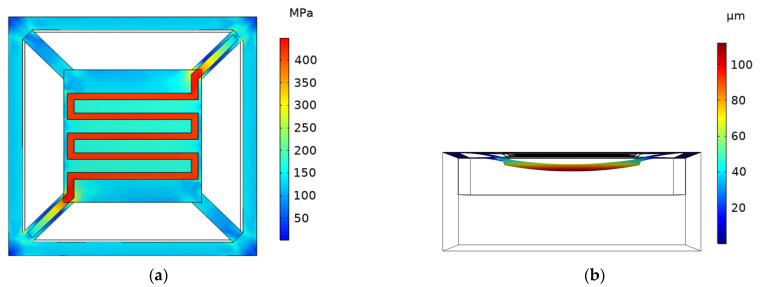
Cloud diagram of stress, and deformation displacement simulation of CCLC model. (**a**) Cloud diagram of stress simulation; (**b**) cloud diagram of deformation displacement simulation.

**Figure 5 micromachines-13-02252-f005:**
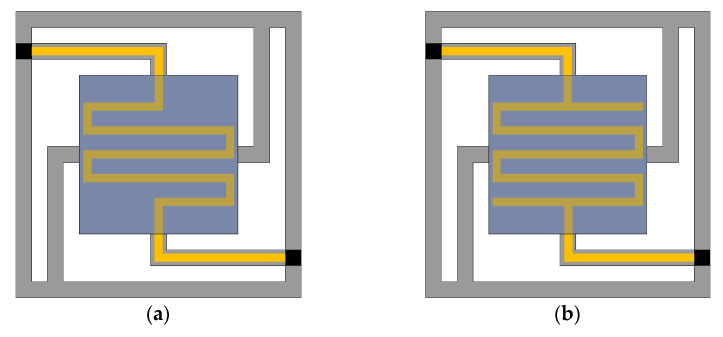
Tow ECLC models. (**a**) I-ECLC model; (**b**) II-ECLC model.

**Figure 6 micromachines-13-02252-f006:**
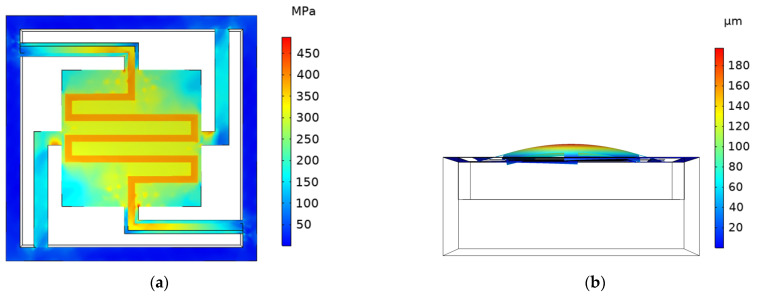
Cloud diagram of stress and deformation displacement simulation of I-ECLC model. (**a**) Cloud diagram of stress simulation; (**b**) cloud diagram of deformation displacement simulation.

**Figure 7 micromachines-13-02252-f007:**
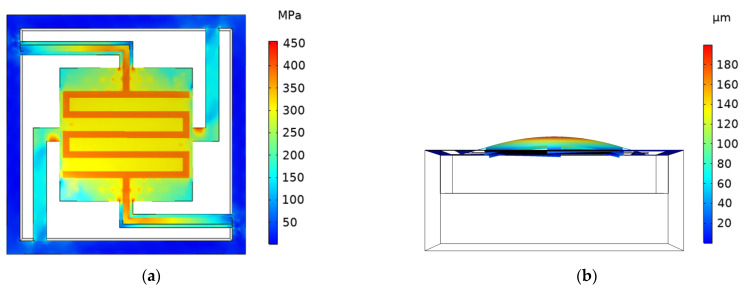
Cloud diagram of stress and deformation displacement simulation of Ⅱ-ECLC model. (**a**) Cloud diagram of stress simulation; (**b**) cloud diagram of deformation and displacement simulation.

**Figure 8 micromachines-13-02252-f008:**
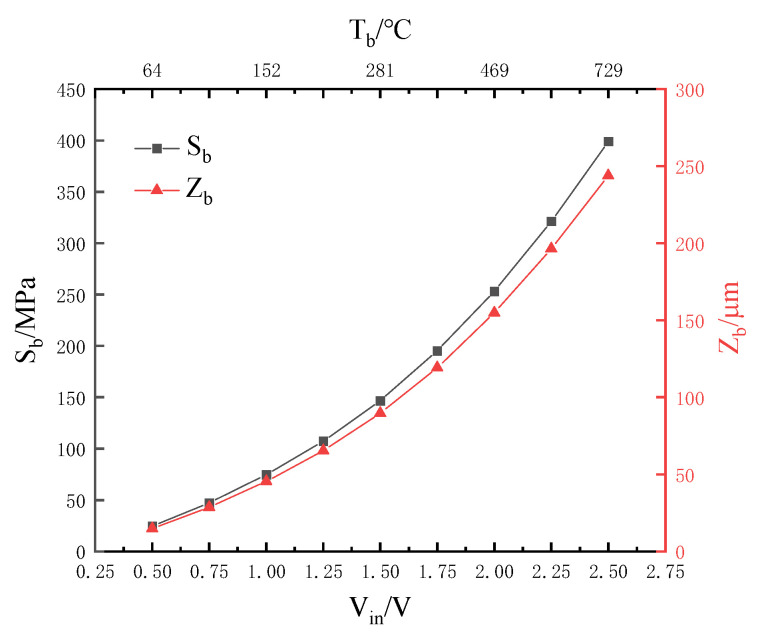
$T_{b}$, $S_{b}$, and$\;Z_{b}$ under different $V_{in}$ in II-ECLC model.

**Figure 9 micromachines-13-02252-f009:**
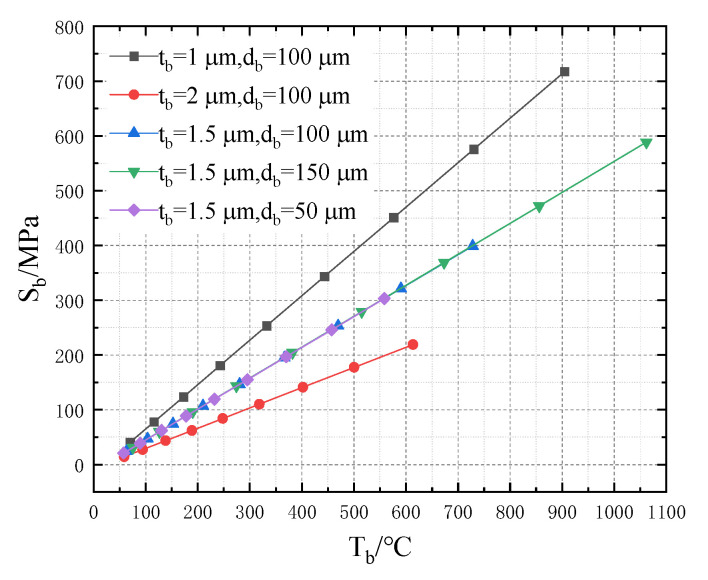
Relationship between $S_{b}$ and $T_{b}$.

**Figure 10 micromachines-13-02252-f010:**
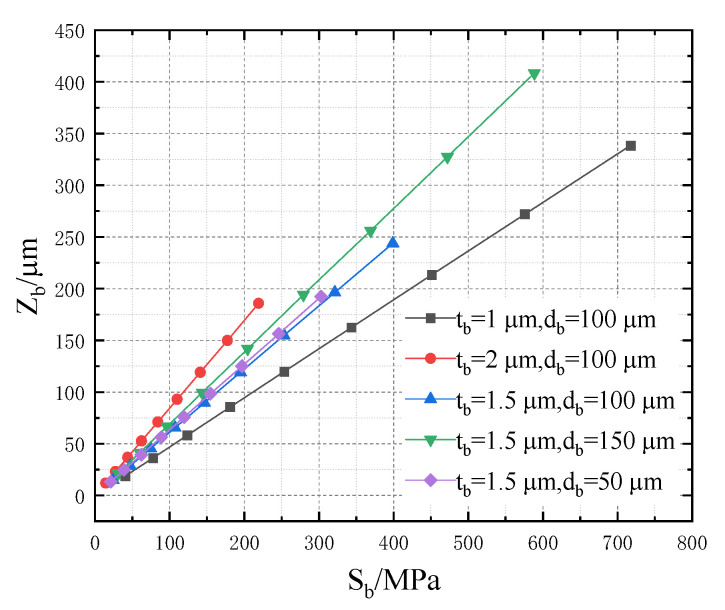
Relationship between $Z_{b}$ and $S_{b}$.

**Figure 11 micromachines-13-02252-f011:**
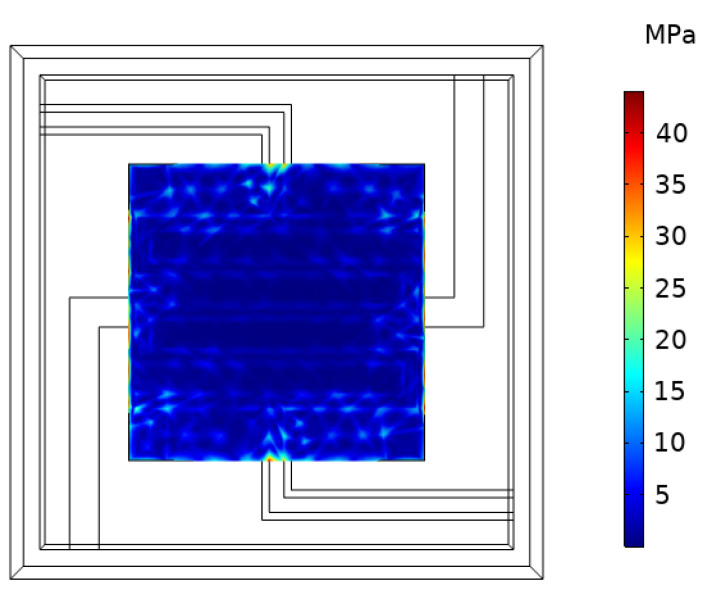
Cloud diagram of interfacial stress simulation between insulating layer and getter.

**Figure 12 micromachines-13-02252-f012:**
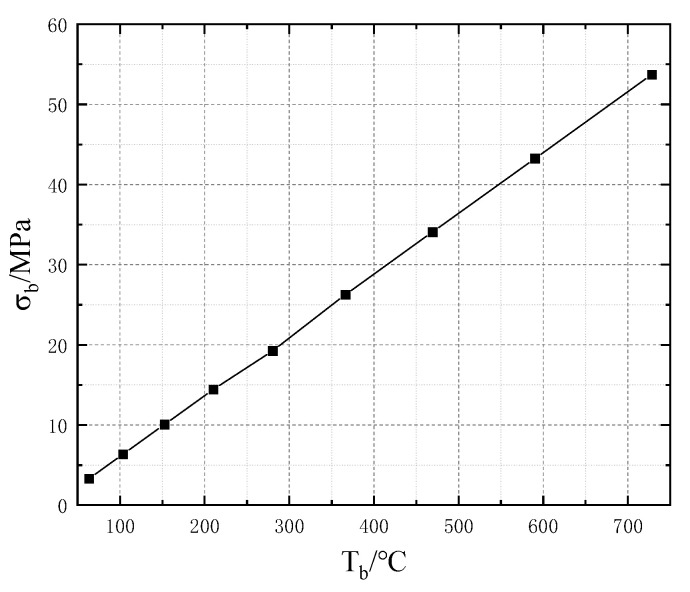
Relationship between $\sigma_{b}$ and $T_{b}$.

**Table 1 micromachines-13-02252-t001:** Floating structure parameters.

Structure	Material	Dimensions [μm]	Thermal Conductivity [W/m·K]	Thermal Expansion Coefficient [1/K]
Substrate	Silicon	1800 × 1800 × 700	131	2.33 × 10^−6^
Cavity	…	300 × 50 (*t_c_* × *d_c_*)	0.01	…
Cantilever structure	Silicon nitride	1.5 × 50 × 1150 (*t_b_* × *d_b_* × *l_b_*)	20	2.3 × 10^−6^
Heater	Titanium	0.2 × 50 × 100 (*t_m_* × *w_1_* × *w_2_*)	21.9	8.6 × 10^−6^
Insulating layer	Silicon nitride	0.4 (*t_s_*)	20	2.3 × 10^−6^
Getter	Titanium-based alloy	3 (*t_g_*)	21.9	8.6 × 10^−6^

## Data Availability

Not applicable.

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
