# Peer review of "Structural Optimization and Mechanical Simulation of MEMS Thin-Film Getter–Heater Unit"

_micromachines, 2022, doi:10.3390/mi13122252_

Round 1
Reviewer 1 Report
In this study,the floating structure model of the thin-film getter–heater unit has been deeply analyzed based on the finite element method in order to avoid structure breakage or too large deformation and film delamination by thermal stress at high temperatures. Several structures were established for simulation, and found that edge-center-located cantilever model has smaller thermal deformation and an upward deformation trend, compared with the edge- edge-located cantilever model. I think this study is sysmatically show how to design a optmized MEMS sensor, which can be accepted by Micromachines .
Reviewer 2 Report
Previous, the author published a paper for a micro-thin-film getter-heater unit for high vacuum capsulation of MEMS devices. In this paper, structure optimization based on COMSOL simulation was presented to avoid failure due to high stress. The paper may be published after the followings could be addressed.
1. What is the mesh size of the simulation? The design looks have multiple 90 degree corners. Did the author round the corner? If not, does the stress at the corner converge as mesh size decreasing?
2. Fig 1 shows the 2D diagram of the design. It would be better if the author can show us a 3D model.
3. Major dimensions (beam/block/gap width, lenght, etc.) of the design are not presented in the paper. Please present the major dimension of each design and justify why these dimensions are chosen.
4. The isolation layer is claimed as Nitride, which could be any of a class of chemical compounds of nitrogen where nitrogen has a formal oxidation state of −3. What specific nitride was used in the design?
Reviewer 3 Report
Generally speaking, the manuscript is interesting and written well. Some symbols should be italic.
Two questions:
1. Why does there exist speckles in the could diagrams?
2. How well would the fabrication process of the floating structure be compatible with other processes of the targeted devices?
Reviewer 4 Report
The authors successfully described the method of design of a MEMS cantilever structure in high temperature. I think the manuscript was well written. However, I would ask the authors to respond to the comments.
In general, SiN has residual stress. The value depends on the method of deposition. I think the stress affects the result of your calculation. I would like to confirm the authors have taken into account the residual stress of SiN.
p.2, L 50 “SAEA getter” seems to be “SAES getter”.
p.2, Please put explanations of tb and db in the text.
p.8, 9 Coefficients in Eq.(14,15,18) seem to be error.
